# What Motives Influence Parents’ Commitment to Their Children’s Sport Participation in the United States?

**DOI:** 10.3390/ijerph22101473

**Published:** 2025-09-24

**Authors:** Katherine N. Alexander, Daniel J. M. Fleming, Mitchell Olsen, Travis E. Dorsch, Kat V. Adams

**Affiliations:** 1Department of Human Development and Family Studies, Utah State University, Logan, UT 84341, USAtravis.dorsch@usu.edu (T.E.D.); 2School of Sport, Exercise, and Rehabilitation, University of Hull, Hull HU6 7RX, UK; dan.fleming@hull.ac.uk; 3School of Kinesiology and Nutrition, University of Southern Mississippi, Hattiesburg, MS 39406, USA; kat.adams@usm.edu

**Keywords:** sport parenting, motivation, values, sport parent decision-making

## Abstract

**Background**: The public often places value on youth sport involvement in the United States due to its potential to foster positive outcomes for participants. Although sport parents are key socializers and provide access to appropriate participation opportunities for children, less is known about how their perceptions of their child’s motives influence their sport commitments. **Purpose**: Therefore, the purpose of the present study was to understand how parents’ perceptions of their child’s motives for sport participation were associated with time/travel sport commitments. **Methods**: Participants (N = 1250) were parents in the United States reporting on their child’s youth sport participation. Measures assessed their perceptions of their child’s motives for sport involvement, how many hours per week and months per year they engaged in sport, and how far they tended to drive to facilitate sport opportunities. Multiple regressions were utilized. **Results**: Analyses revealed that the number of months per year was positively predicted by motives for being physically healthy and spending time with friends. Similarly, being with friends was a positive predictor of the number of weekly hours spent in organized sport and having fun positively predicted the distance driven to participate. Motives for becoming more physically attractive negatively predicted time and travel commitments. **Conclusions**: Overall, the present study sheds light on how the ways parents perceive their children’s motivations for participating in youth sport influences parents’ commitment to facilitating sport participation opportunities for their children.

## 1. Introduction

Youth sport participation is a public health issue because sport can promote increased physical activity and positive developmental outcomes. Moreover, many people participate in this setting, with at least half of youth in the United States having been involved in organized sport at some point during their development [1]. This highlights a strong possible avenue for youth-focused policy and programming for physical activity and socioemotional development, yet there is a need to better understand what makes families and children continue to be involved in sport. Numerous motivations for this continued engagement have been reported, including improving athletic capabilities, enjoyment, learning new skills, socializing, appeasing parents, and weight or appearance management being common reasons why these athletes may continue to be involved in sports [2,3,4]. Moreover, parents play an especially prominent role in shaping their child’s engagement in this setting and serve as primary advocates, key socializing agents, providers and interpreters of experiences, models of appropriate behaviors, teachers, and reinforcers of specific values in sport [5,6,7].They must cater to their child’s needs and values to facilitate positive sport experiences [4,5], yet few studies quantitatively examine how parental perceptions of their child’s values may differentially be associated with their commitment to provide time and access to sport [5,7].

Parental perceptions of their child’s motives and abilities influence how they invest in their child’s youth sport participation. These caregivers play a substantial role in their child’s sport participation as a primary supplier of opportunities [5], whilst managing a range of individual and environmental factors that influence this provision (e.g., individual family financial constraints, broader societal sport trends) [8]. They are expected to make decisions that are in the best interests of their child, but this is not a simple or innate skill. Qualitative research suggests that parents have conflicting goals around their child’s sport involvement and often simultaneously display pressuring and supporting behaviors at practices and competitions [9,10]. Incongruence between parent- and child-reports for the same measure is also common in the literature, though very few studies examine this incongruence [10] may indicate differing perspectives and understandings of sport experiences, which could impact parent’s ability to provide appropriate opportunities for their child to achieve their sporting goals [11]. Regardless, expert sport parents can navigate these complexities and challenges to skillfully select participation opportunities or to provide tangible support to their child, which often include transportation or time investments [11].

Regardless of child perceptions, parents have comparatively more power in sport decision-making for their child. Qualitative studies indicate that parental involvement in and commitments to their child’s sport are influenced by numerous factors, including parents’ own goals and beliefs about sport [12], Despite the dissonance between the sport-related goals they hold for themselves about their child and their child’s own sport-related goals [9], parents often have the ultimate power in the family to select what investments are made related to their child’s youth sport participation. Scholars highlight that expert sport parents are effective at consolidating incongruent and conflicting goals to identify and provide an appropriate context for their child [11], and perceptions of child motives are especially relevant to the intensity of familial investments. However, limited quantitative research has been designed to examine how parental perceptions of their child’s sport motives are associated with time and other familial commitments, so the relation of these perceptions with commitments is largely unknown.

### 1.1. Eccles’ Expectancy-Value Model as a Guiding Interpretative Framework

Eccles’ expectancy-value model [13] provides a potentially strong interpretive framework for understanding these complex relationships across child motives and abilities, parent’s perceptions surrounding their child’s motives and abilities, and subsequent sport commitments. It has been applied to academic and sport domains, and the model suggests that child sport aptitudes, expectations for success, and subjective task values (i.e., incentive value, attainment value, utility value, and cost) are related to achievement choices of youth athletes [14].The model also includes relevant socializing agents that directly and indirectly influence sport participation behaviors (e.g., directly via providing opportunities and expressing certain beliefs or behaviors, indirectly via child’s understandings of socializers’ beliefs and expectations) [6,14].

Through the lens of Eccles’ expectancy-value model, parents serve as an especially salient socializer and may reinforce or inhibit their child’s sport investments through both implicit and explicit means. Implicitly, parents’ perceptions of their children’s motives and sport aptitudes influence how children understand their own experiences, including their expectancies for success and perceived values inherent to sport [15]. Explicitly, parents’ decision-making around sport commitments directly shapes their child’s opportunities for involvement and success. Parents also have more power in making decisions for their child, meaning that their perceptions are crucial to understanding how they reinforce or inhibit their child’s sport opportunities. Given this important role that parents play [5,6,7,11,16], there is a need to better understand quantitatively how parental perceptions of their child’s sport motives are associated with familial commitments. However, no studies to-date explore parental perceptions of such motives and their subsequent support or inhibition of sporting opportunities. Understanding this association would ultimately provide more insights into which child goals parents believe are valuable in youth sport settings, and this has implications for how they invest in their child’s youth sport participation.

### 1.2. The Current Study

The current study utilized a quantitative and cross-sectional approach to explore the associations of parental perceptions of their child’s reasons for sport involvement and subsequent familial time and travel sport commitments. Participants were currently limited to those residing in four states in the Intermountain West region of the United States given original aims of the grant-funded research project and dataset. The following research question guided the choice of measures and analytic procedures: how do parents perceive their children’s motivations for participating in youth sport, and how those motivations influence their commitment to providing sport participation opportunities? No predetermined hypotheses were proposed due to a lack of supporting research literature and the exploratory nature of the research question.

## 2. Materials and Methods

### 2.1. Participants

Participants (N = 1250) were parents of youth sport participants residing in four states in the Intermountain West region of the United States given the broader aims of the research project. The sample is a subset of a broader study examining organized youth sport, unstructured physical activity, and land-use recreation in this region. Participants were included in the present sample if they indicated that their child was taking part in at least one hour of organized youth sport per week. Parents were, on average, 37.27 years old (SD = 8.91) and reported a mean household income of $106,279.26 (SD = 165,869.88). A majority (58.6%) identified as male, with 39.5% identifying as female, 1.3% as non-binary, and 0.5% as “other” or “prefer not to say.” Most identified as White (70.4%), followed by Hispanic, Latino, or Spanish origin (14.9%), Black or African American (7.6%), Asian (3.4%), American Indian or Alaskan Native (1.1%), Multiracial (0.6%), Native Hawaiian or Pacific Islander (0.6%) and “other” or “prefer not to say” (1.0%). This racial/ethnic composition is representative of the region (U.S. Census, 2022). More than half (61.5%) of parents in the sample were married, with 17.0% identifying as single, never married, 9.1% as living with their partner but not married, 7.0% as divorced, 2.2% as widowed, and 1.8% as separated. The remainder (1.2%) identified their relationship status as “other” or “prefer not to say.” Most parents (68.6%) reported being employed full time, 7.6% as being employed part time, 7.4% as a homemaker, 5.9% as being self-employed, 2.4% as unable to work, 2.9% as students, and 2.0% as “other” or “prefer not to say,” 1.9% as out of work, and 1.3% as retired. Finally, parents identified their community type as urban (46.7%), suburban (34.6%), and rural (18.6%).

### 2.2. Procedure

The study was approved by the Utah State University Institutional Review Board and adhered to APA-7th edition ethical standards. Participants were recruited from across the United States via paid Qualtrics panel. Participants provided informed consent and completed the online questionnaire were screened by a Qualtrics survey management team based on pre-established quotas for participant sociodemographic characteristics as well as the completeness of their responses. Participants who did not meet the inclusion criteria (i.e., frequent interactions with at least one child who has participated in sport and/or physical activity and being at least 18 years of age), and those who only partially completed surveys were excluded from the analysis. Additionally, measures were taken during data collection and data screening to limit the infiltration of bots in the survey sample, including not directly advertising incentives immediately and having online checks in-place. The final sample (N = 1250) comprised parents who met the inclusion criteria and who had children actively participating in at least one organized youth sport. Data were collected between 25 July and 7 November 2022.

### 2.3. Measures

#### 2.3.1. Sport Participation Motivation

A single, study-designed item was utilized to assess parents’ perceptions of the target child’s motivation for engaging in youth sport activities. The item response set included “to be physically healthy,” “to become more physically attractive,” “to improve at an activity,” “to be with friends,” and “to have fun.” Parents indicated how important they perceived these motivations to be for their child on a Likert scale ranging from 1 (not at all important) to 5 (very important). This design choice was utilized to reduce participant fatigue and to capture simple responses to the question of motives.

#### 2.3.2. Commitment to Sport

Parents were asked to indicate how many months a year and how many hours per week their child participates in organized youth sport, along with, on average, how many miles they drive to afford their child this opportunity. Weekly hours were reported for three domains: formal practice, independent practice, and competition. The responses for these three items were summed to indicate the total number of hours per week youth were engaged in their organized sport endeavors.

### 2.4. Data Analysis

Statistical analyses were conducted using R [17]. Descriptive statistics and correlations were examined based on the recommendations of [18], using the psych [19], and apaTables [20] packages for R. To answer our research questions, we fit three multiple regression models, each containing one of the three dependent variables (months per year, hours per week, and distance driven). All five independent variables were entered in one step, as the change in R^2^ related to model testing was not identified as a key statistic in the project.

## 3. Results

### 3.1. Descriptive Statistics

Descriptive statistics and correlations were calculated for all five independent and all three dependent variables. The results of these analyses can be found in Table 1 and Table 2. Results highlight small to moderate correlations between all research variables.

### 3.2. Multiple Regression Analyses

The first multiple regression model fit included the number of months per year that a child participated in youth sport as the dependent variable. As identified in the descriptives, the mean was 5.85 months per year (*SD* = 3.10). Results of the multiple regression identified three significant predictors of this. Positive relationships were identified between “to be physically healthy” (*b* = 0.35, 95% CI [0.14–0.56], *SE* = 0.11, *p* = 0.001) and “to be with friends” (*b* = 0.31, 95% CI [0.09–0.52], *SE* = 11, *p* = 0.005). Perceived motives related to “to become more physically attractive” was a significant negative predictor (*b* = −0.28, 95% CI [−0.42–−0.14], *SE* = 0.07, *p* < 0.001). Motives “to have fun” and “to improve at an activity” were non-significant predictors in the regression model. These five independent variables accounted for only 4.8% of the variance in the number of months per year that children participated in youth sport. In sum, children participated in more months throughout the year if parents believed youth valued sport for physical health or for friendships. However, they participated in fewer months per year of sport if being attractive was perceived as valued by their child. The full table of results are displayed in Table 3 and Table 4.

The second model included the number of hours per week that children participated in youth sport as the dependent variable. The mean score was 14.53 h per week (*SD* = 0.95). There was one significant relationship identified in this model, indicating that “to be with friends” was the sole predictor of weekly hours in this instance (*b* = 1.59, 95% CI [0.46–2.71], *SE* = 0.73, *p* = 0.006). Overall, this model accounted for 1.1% of the variance in the number of weekly hours in the youth sport context. As such, children participated for more hours each week if their parent valued them spending time with their friends. The full results are available in Table 4.

The third model included the number of miles driven to participate in youth sport as the dependent variable. On average, parents reported driving 12.89 miles (*SD* = 20.61) to have their child participate in organized youth sport. There were two significant relationships identified in this model. Similarly to the first model, “to become more physically attractive” was a negative predictor of the distance driven (*b* = −1.37, 95% CI [−2.28–−0.46], *SE* = 0.46, *p* = 0.003). The perceived motive “to have fun” was a significant positive predictor (*b* = 1.90, 95% CI [0.31–3.49], *SE* = 0.81, *p* = 0.019). In sum, this model accounted for 1.9% of the variance in the number of miles parents drive to have their children participate in youth sport. To summarize, parents drove further to have their children participate in youth sport if they felt that their children participated to have fun, while they drove less if they believed that their child participated to become more physically attractive. The full results are available in Table 5. Notably, the variance explained across all models was very low, ranging from 0.011 to 0.048.

## 4. Discussion

The current study utilized a quantitative and cross-sectional approach to explore the associations of parental perceptions of their child’s motivations for sport involvement and subsequent familial time and travel sport commitments. Parents were asked to assess what motives were pertinent to their child’s desire to participate in sport involvement (“to be physically healthy,” “to become more physically attractive,” “to improve at an activity,” “to be with friends,” and “to have fun.”) and reported on how often their child participates in youth sport and how far they drive to provide this participation. Findings highlighted that parents that perceived that their child participated in sport to be physically healthy and spend time with friends had increased time commitments in terms of months spent across the year. Spending time with friends was also associated with increased time spent weekly on organized youth sport involvement. Parents who believed that their child participated to have fun reported increased travel commitments related to driving further to provide youth sport opportunities. Perceived child motives related to improving at an activity were not significantly related to parent commitments in terms of monthly or weekly time in sport or travel distances. Interestingly, parents who believed that their child valued becoming more physically attractive as a major motive for sport involvement reported reduced investments (i.e., spending fewer months and traveling less).

### 4.1. Perceived Child Values That Lead to Increased Investments

Findings highlighted that parents were more likely to invest in organized youth sport when they believed that their child had motives for participation related to being physically healthy, being with friends, and having fun. This is in alignment with research that examines youth sport as a setting for positive physical, emotional, and social development [22,23]. Indeed, existing work has also suggested that parents enjoy these experiences alongside their child and gain parental satisfaction from observing their child have fun and enjoy playing [2,5].

In contrast, the non-significant associations of perceived motives around improving skills and time and travel commitments do not fully align with current research. Specifically, athlete-reported motivations and other purported benefits of organized youth sport involvement regularly list competence and enhanced performance capabilities as significant reasons for continued sport involvement [2,3,24]. Results from the current study might indicate that improvement is not a substantial enough reason for parents and families to commit to organized youth sport participation. Findings could alternatively be indicative of the unique and complex role that parental perceptions of their child’s reasons for sport involvement play on the intensity of familial commitments [24]. Future research could consider the roles of perceived child motives, parental motivations for their child’s sport involvement, and resulting commitments to better understand what values are most important to children, parents, and families.

### 4.2. Perceived Child Values That Lead to Decreased Investments

In the current study, parents who perceived that their child was motivated to participate in sport to become more physically attractive were less likely to commit more time across the year or to travel farther distances to facilitate their child’s youth sport participation. This finding is novel since it concerns parental understandings of child motives, and there is no pre-existing literature that would explain these relations. Further, the specific impact of parents in shaping positive and negative sport motives and behaviors of youth athletes related to weight and appearance management is relatively understudied in the current literature, with select scholars examining aspects of weight-related maltreatment as it concerns parents and athlete entourages, see [25] for an example article. Future work is needed to corroborate findings in the current study.

### 4.3. Eccles’ Expectancy-Value Model in Explaining Alternative Factors

Given that the variance explained (R^2^) values were low in the current study, parent perceptions of child motives may not play a dominant role in how they make sport commitments to facilitate their child’s participation. Instead, other factors may be more important, including access to resources, broader family commitments beyond the sporting domain, or other environmental or social factors [8]. Eccles’ expectancy-value model might suggest that child perceptions around sport, incongruence or congruence between parent and athlete perceptions, or other socializer-based factors shape sport experiences and commitments. Examining these factors may subsequently explain more variance statistically in understanding parental commitments and may explain such behaviors more directly and practically. Moreover, sport parents could rely more on contextual factors (e.g., cultural stereotypes, family demographics, family structure and siblings), perceptions of the child’s sport aptitude, or perceived values apparent in sport when providing opportunities to their child [14,15]. Eccles’ expectancy-value model subsequently provides more avenues for understanding factors that shape parental commitments in sport, and future works could utilize this framework to better understand how parental beliefs, behaviors, and values are connected to child experiences and outcomes within sport settings.

### 4.4. Practical Implications

Several practical implications are apparent from the current study. Given that youth sport participation is a public health issue and a major source of physical activity for many children and adolescents [1], findings can be utilized to facilitate increased involvement. Policies and programs should explicitly target specific health and socioemotional outcomes [26], rather than focusing solely on sport-specific skills or competition. Furthermore, findings highlighted that parental perceptions played only a modest role in shaping youth sport investments compared to other factors; more research is needed to understand what perceptions, socializer behaviors, and contexts play a more prominent role in shaping familial commitments to youth sport. Understanding this broader array of cognitions and determinants on youth sport commitments would allow for more effective and focused interventions and policies that keep youth and their families physically active and engaged in sport.

Those working in organized youth sport settings (e.g., coaches, administrators) should be aware of how parent and child motives shape investments in sport. Both families and children value sport for the development of sport-specific and life skills, and robust programming around implicit and explicit ways to promote positive youth development in sport are available [26]. Coaches and program designers might specifically focus on programming that promotes psychosocial development, security, mentorship, positive relationships, and autonomy support [27]. Additionally, educational programs that facilitate positive and assertive communication between parents and children could reduce incongruencies in perceptions and expectations in sport and could serve to ensure that sport parents act as experts in providing appropriate and positive opportunities for their child [7,11].

### 4.5. Limitations and Future Directions

Despite the contributions of the present study, certain limitations of the research should be acknowledged. Findings are limited to the context within which the data were collected. In this case, data are cross-sectional, and participants were youth sport parents of athletes in the Intermountain West region of the United States. As such, the findings of the present study are unable to speak to temporal relationships between variables and may not be generalizable to broader United States, or Western, contexts. Future work may be designed to collect a longitudinal, nationally representative sample, or even multi-national sample, to assess these relationships across time and contexts.

In addition, while this project primarily focused on parental perspectives, it would be beneficial to also utilize children’s experiences to better understand their motivations for participating in youth sport and to understand child and parent alignment in interpretations. Although parents serve as the ultimate decision-makers in facilitating sport participation, it is evident from prior research that they are not particularly good at aligning their goals with that of their children [10]. This means that there is a possibility of reporting bias and incongruencies, especially since parental perceptions might not align with child’s perceptions and motivations. It is thus important to understand the child’s motivations and their own perceptions related to their caregivers to capture more holistic insights into their experiences of sport. Future work could be strategically designed to explore participation motivation from children’s perspectives and to compare alignment between parents and children.

Furthermore, the current study relies on single-item indicators to examine perceptions of child motives. Employment of these indicators was useful in reducing participant burden in the recruitment phase of the research project and in simply capturing measures of specific perceptions related to the question of what specific motives they believed were pertinent to their child’s desire to participate in sport involvement (i.e., “to be physically healthy,” “to become more physically attractive,” “to improve at an activity,” “to be with friends,” and “to have fun”). However, future research should consider alternative multi-item indicators and broader assessments when understanding parent perceptions of their child’s motivations.

## 5. Conclusions

Parents have been identified as one of the primary social agents involved in providing a positive youth sport experience [5,16]. However, they are not always attuned to their children’s motivations for participating in youth sport [10,16]. In their fiduciary role as the primary provider of youth sport experiences, parents are expected to provide suitable opportunities that align with their children’s motivations for participation. The present study extends the youth sport literature by exploring how parents’ perceptions of their children’s youth sport motivations are related to their tangible investment in youth sport, as measured by three tangible aspects of their involvement (the number of months per year their child participates, the hours per week they are engaged, and the distance they drive to facilitate their child’s participation in organized sport. Findings suggest that if parents perceive their children want to have fun, spend time with friends, and be active then they enabled youth sport participation. However, if they perceived that their children wished to take part simply to be attractive, then they would provide less youth sport opportunities. Given that variance explained was low across all models, parental perceptions evidently play only a modest role in shaping youth sport investments compared to other factors and determinants. These findings add to prior research by identifying that these trends persist in parent’s provision of youth sport opportunities as well as existing within individual participants. Accordingly, parent expertise in encouraging positive engagement in youth sport may begin with the transmission of positive values and motivations relating to the sport context.

## Figures and Tables

**Table 1 ijerph-22-01473-t001:** Means and Standard Deviations.

Variable	*M*	*SD*
**1. Months per year**	5.85	3.10
**2. To be physically healthy**	4.19	1.04
**3. To become more physically attractive**	3.55	1.37
**4. To be with friends**	4.23	0.97
**5. To have fun**	4.41	0.86
**6. To improve**	4.18	0.95
**7. Weekly hours**	14.53	16.07
**8. Drive distance**	12.89	20.61

*Note*. *M* and *SD* are used to represent mean and standard deviation, respectively.

**Table 2 ijerph-22-01473-t002:** Correlations with Confidence Intervals.

	1	2	3	4	5	6	7
**1.**							
**2.**	0.15 ** [0.09, 0.20]						
**3.**	−0.04 [−0.10, 0.01]	0.35 ** [0.30, 0.39]					
**4.**	0.14 ** [0.09, 0.19]	0.46 ** [0.41, 0.50]	0.30 ** [0.24, 0.35]				
**5.**	0.13 ** [0.07, 0.18]	0.41 ** [0.36, 0.45]	0.10 ** [0.04, 0.15]	0.45 ** [0.41, 0.50]			
**6.**	0.12 ** [0.06, 0.17]	0.57 ** [0.53, 0.60]	0.30 ** [0.25, 0.35]	0.45 ** [0.41, 0.50]	0.48 ** [0.44, 0.52]		
**7.**	0.15 ** [0.10, 0.21]	0.01 [−0.04, 0.07]	0.05 [−0.00, 0.11]	0.07 ** [0.02, 0.13]	−0.01 [−0.07, 0.04]	0.04 [−0.02, 0.09]	
**8.**	0.20 ** [0.14, 0.25]	0.07 * [0.01, 0.12]	−0.06 * [−0.11, −0.00]	0.06 * [0.00, 0.11]	0.10 ** [0.05, 0.16]	0.05 [−0.00, 0.11]	0.11 ** [0.05, 0.16]

*Note*. Numbers correspond to variables in Table 1. Values in square brackets indicate the 95% confidence interval for each correlation. The confidence interval is a plausible range of population correlations that could have caused the sample correlation [21]. * indicates *p* < 0.05. ** indicates *p* < 0.01.

**Table 3 ijerph-22-01473-t003:** Regression Results with Months Per Year as the Criterion.

Predictor	*b*	95% CI [LL, UL]	*se*	*t*	*r*	Fit
(Intercept)	2.89 **	[1.88, 3.91]	0.52	5.61		
To be physically healthy	0.35 **	[0.14, 0.56]	0.11	3.27	0.16 **	
To become more physically attractive	−0.28 **	[−0.42, −0.14]	0.07	−4.04	−0.04	
To be with friends	0.31 **	[0.09, 0.52]	0.11	2.82	0.15 **	
To have fun	0.16	[−0.07, 0.40]	0.12	1.36	0.14 **	
To improve at an activity	0.10	[−0.13, 0.33]	0.12	0.85	0.13 **	
						*R*^2^ = 0.048 **
						95% CI [0.02, 0.07]

*Note.* A significant *b*-weight indicates the beta-weight and semi-partial correlation are also significant. *b* represents unstandardized regression weights *r* represents the zero-order correlation. *LL* and *UL* indicate the lower and upper limits of a confidence interval, respectively. ** indicates *p* < 0.01.

**Table 4 ijerph-22-01473-t004:** Regression Results with Weekly Hours as the Criterion.

Predictor	*b*	95% CI [LL, UL]	se	t	*r*	Fit
(Intercept)	10.73 **	[5.40, 16.05]	2.71	3.95		
To be physically healthy	−0.51	[−1.62, 0.60]	0.57	−0.91	0.02	
To become more physically attractive	0.39	[−0.33, 1.11]	0.37	1.06	0.06 *	
To be with friends	1.59 **	[0.46, 2.71]	0.57	2.76	0.08 **	
To have fun	−1.07	[−2.32, 0.18]	0.64	−1.69	−0.01	
To improve at an activity	0.61	[−0.62, 1.84]	0.63	0.98	0.04	
						*R*^2^ = 0.011 *
						95% CI [0.00, 0.02]

*Note.* A significant *b*-weight indicates the beta-weight and semi-partial correlation are also significant; *r* represents the zero-order correlation. LL and UL indicate the lower and upper limits of a confidence interval, respectively. * indicates *p* < 0.05. ** indicates *p* < 0.01.

**Table 5 ijerph-22-01473-t005:** Regression Results with Drive Distance as the Criterion.

Predictor	*b*	*b* 95% CI [LL, UL]	*se*	*t*	*r*	Fit
(Intercept)	2.32	[−4.44, 9.09]	3.44	0.67		
To be physically healthy	1.22	[−0.19, 2.63]	0.72	1.70	0.07 *	
To become more physically attractive	−1.37 **	[−2.28, −0.46]	0.46	−2.94	−0.05	
To be with friends	0.53	[−0.90, 1.96]	0.73	0.73	0.06 *	
To have fun	1.90 *	[0.31, 3.49]	0.81	2.35	0.11 **	
To improve at an activity	−0.09	[−1.65, 1.47]	0.80	−0.11	0.05	
						*R*^2^ = 0.019 **
						95% CI [0.00, 0.03]

*Note.* A significant *b*-weight indicates the beta-weight and semi-partial correlation are also significant. *r* represents the zero-order correlation. LL and UL indicate the lower and upper limits of a confidence interval, respectively. * indicates *p* < 0.05. ** indicates *p* < 0.01.

## Data Availability

Data is available by contacting the author Dr. Travis Dorsch at travis.dorsch@usu.edu.

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
