# Peer review of "What Motives Influence Parents’ Commitment to Their Children’s Sport Participation in the United States?"

_ijerph, 2025, doi:10.3390/ijerph22101473_

Round 1

Reviewer 1 Report

Comments and Suggestions for Authors

see attachment

Author Response

Reviewer 1

This manuscript addresses how parents’ perceptions of their children’s motives for participating in sports are associated with sport-related time and travel commitments. The topic is relevant and timely, as it is key to determining the extent and nature of parental support, and it also allows for the identification of possible incongruences with children’s actual motivations. However, several important issues in the introduction, method, and discussion sections need to be addressed before the manuscript can be accepted for publication.

  1. Introduction

The introduction is overly circular, repeatedly emphasizing the role of parents and their perceptions of children’s motivations without adding new nuances, and it does not clearly delineate the knowledge gap.

The authors added more information throughout the introduction to highlight that most of the current knowledge is qualitative and not quantitative in nature. Information on Eccles’ expectancy-value model was additionally added to highlight nuances and theorized relationships of variables. See subsection 1.1 on Page 2 for an example.

Lines 42–45 discuss the study purpose:

“The current study utilized a quantitative approach and was designed to explore associations between different parental perceptions of their child’s reasons for sport involvement and subsequent time and travel commitments related to sport.” This statement is repeated again in lines 80–82: “The current study utilized a quantitative and cross-sectional approach to explore the associations of parental perceptions of their child’s reasons for sport involvement and

subsequent familial time and travel sport commitments.” It is recommended to remove lines 42–45.

The authors deleted the statement at lines 42-45 based on reviewer recommendations. See page 2, lines 42-45.

In line 56, there is a citation error that does not follow the journal’s style guidelines.

The authors changed this citation to meet journal style guidelines.  See page 2, line 59.

The idea that parents’ perceptions of their children’s motivations influence their investments of time and resources is repeated across different paragraphs with almost the same wording, without sufficiently deepening the problem. It is suggested to clearly describe the parental role and what is already known about their perceptions. The knowledge gap (what is not known and why it is important to know) should be specified in more detail.

The authors have added several contextual sentences and information on the theoretical framework to address these concerns on repetition and knowledge gaps. See the first full paragraph on page 2, starting with lines 47-48, for an example.

  1. Method

The sample, procedure, and instrument are well described.

  1. Results

The results are clearly explained.

The authors thank the reviewer for this positive feedback on the method/results sections.

  1. Discussion

The low R² values indicate that other factors (economic, logistical, cultural, availability of facilities, number of children, etc.) may weigh more heavily in parental commitment, and these were not measured. This should be discussed. The text cites previous studies on the lack of alignment between parental and child goals but does not connect this finding to the current data.

Low variance explained and the role of parent-child congruence have been implemented into the discussion section of the paper. See Section 4.3 on page 9 for an example.

Finally, the discussion does not explain how the findings could be applied in sports programs or parental involvement strategies. Including this would give greater weight to the study.

A practical implications subsection has been added to the discussion to allow for a more thorough discussion of possible implications. See section 4.4 on page 10 for an example. 

Reviewer 2 Report

Comments and Suggestions for Authors

The manuscript addresses an important topic by examining how parents’ perceptions of their children’s sport motives influence time and travel commitments. The study is based on a large and diverse sample, which is a clear strength. The paper is well written, but several areas require clarification or revision before the work can reach publication quality:

Introduction

Comment 1: The introduction lacks a strong theoretical foundation. Although Eccles’ expectancy–value model is introduced in the Discussion (section 4.3), it should be included earlier as a guiding framework. Even in exploratory research, presenting a conceptual basis helps orient readers and strengthens the study rationale.

Methods

Comment 2:  Children’s motives were measured using single-item indicators. This raises validity and reliability concerns. Please provide stronger justification for this choice and acknowledge it as a limitation. In addition, the sample is restricted to four U.S. states, which limits generalizability. This should be emphasized more explicitly.

Results

Comment 3: The regression models explain very little variance (R² between 1% and 5%). While statistically significant, these findings have limited practical significance. Please highlight this in the Results and discuss the implications more openly.

Discussion

Comment 4: The practical implications are currently underdeveloped. Expand on how coaches, program designers, or parent-education initiatives might use these results.

Comment 5: The interpretation of the negative association between attractiveness motives and parental commitment is speculative. Connect this more firmly to prior literature and avoid over-interpretation.

Comment 6: Since the study relies exclusively on parental perceptions, there is a potential projection bias. Please stress the importance of including children’s voices in future work.

Overall, this study makes a useful contribution, but revisions are necessary to strengthen the theoretical foundation, address methodological limitations, and provide more applied insights.

Author Response

Reviewer 2

The manuscript addresses an important topic by examining how parents’ perceptions of their children’s sport motives influence time and travel commitments. The study is based on a large and diverse sample, which is a clear strength. The paper is well written, but several areas require clarification or revision before the work can reach publication quality:

Introduction

Comment 1: The introduction lacks a strong theoretical foundation. Although Eccles’ expectancy–value model is introduced in the Discussion (section 4.3), it should be included earlier as a guiding framework. Even in exploratory research, presenting a conceptual basis helps orient readers and strengthens the study rationale.

The authors added a section on Eccles’ model in the introduction to better orient readers to the rationale of the present study. See subsection 1.1 on Page 2.

Methods

Comment 2: Children’s motives were measured using single-item indicators. This raises validity and reliability concerns. Please provide stronger justification for this choice and acknowledge it as a limitation. In addition, the sample is restricted to four U.S. states, which limits generalizability. This should be emphasized more explicitly.

The authors added information in the measures and limitations section on use of single-item indicators and additionally added more information on participants being limited to four states in the introduction and methods sections. See page 3 under section 1.2 for an example related to sample limitations and page 11 (lines 383-391) for an example related to indicator limitations.

Results

Comment 3: The regression models explain very little variance (R² between 1% and 5%). While statistically significant, these findings have limited practical significance. Please highlight this in the Results and discuss the implications more openly.

The authors added in information to the results and discussion sections to more thoroughly explain a low variance explained. See page 9 and section 4.3 for an example of this in the discussion section.

Discussion

Comment 4: The practical implications are currently underdeveloped. Expand on how coaches, program designers, or parent-education initiatives might use these results.

A practical implications subsection has been added to the discussion to allow for a more thorough discussion of possible implications. See section 4.4, which starts on page 10.

Comment 5: The interpretation of the negative association between attractiveness motives and parental commitment is speculative. Connect this more firmly to prior literature and avoid over-interpretation.

The authors clarified aspects of previous literature and removed speculations from this subsection in the discussion section. See page 9, section 4.2 for these changes.

Comment 6: Since the study relies exclusively on parental perceptions, there is a potential projection bias. Please stress the importance of including children’s voices in future work.

The authors added more to the limitations section to address this comment by the reviewer. See page 11, lines 378-380 for this change.

Overall, this study makes a useful contribution, but revisions are necessary to strengthen the theoretical foundation, address methodological limitations, and provide more applied insights.

The authors thank this reviewer for their contributions to making the manuscript better. Changes have been made that should ensure that these general points made by the reviewer are all addressed.

Round 2

Reviewer 1 Report

Comments and Suggestions for Authors

Thank you very much for addressing the recommendations

Reviewer 2 Report

Comments and Suggestions for Authors

The manuscript has been substantially improved compared to its previous version. The structure is clearer, the methods and analyses are appropriately reported, and the discussion now better integrates the findings with theory. However, there are still redundancies, minor stylistic issues, and the need for additional conciseness and more recent references.

Introduction

  • Add more recent references (2022–2025) on youth sport parenting and motivation to better anchor the study in current literature.

Methods

  • The use of a single-item measure for motivations is a notable limitation. This is acknowledged later, but it should also be noted in the Methods as a design choice.

Results

  • Please emphasize at the end of the Results section that the variance explained is very low (R² = .048, .011, .019). This contextualization will strengthen the interpretation.

Discussion

  • The discussion is much improved, but:
    • The section on the “physical attractiveness” motive is interesting but speculative and somewhat long. Please condense and anchor it more firmly in cited literature.
    • Section 4.3 (Eccles’ model) repeats concepts already presented in the Introduction. A more concise summary would suffice.
    • Practical implications (4.4) are helpful but could be made more concrete (e.g., specific recommendations for coaches or program designers).

Limitations and Future Directions

  • Well addressed overall. It would be useful to also stress the issue of possible reporting bias, since parents’ perceptions may not align with children’s actual motivations.

Conclusions

  • Clear and concise. However, it would strengthen the take-home message to explicitly state that parental perceptions play a role, but only a modest one, compared to other potential determinants of youth sport engagement.

References

  • The reference list is extensive, but most sources are pre-2020. Please include few recent works (2022–2025) to update the literature base.

Overall, this is a well-designed and clearly written study that contributes novel quantitative insights into how parental perceptions of children’s sport motives shape time and travel commitments. With minor revisions to improve conciseness, and update references, the manuscript will be ready for publication.
